# Risk of Lactic Acidosis in Hospitalized Diabetic Patients Prescribed Biguanides in Japan: A Retrospective Total-Population Cohort Study

**DOI:** 10.3390/ijerph20075300

**Published:** 2023-03-29

**Authors:** Takako Mohri, Sawako Okamoto, Yuichi Nishioka, Tomoya Myojin, Shinichiro Kubo, Tsuneyuki Higashino, Sadanori Okada, Yasuhiro Akai, Tatsuya Noda, Hitoshi Ishii, Tomoaki Imamura

**Affiliations:** 1Department of Diabetes and Endocrinology, Nara Medical University Hospital, Nara 634-8522, Japan; 2Department of Public Health, Health Management and Policy, Nara Medical University, Nara 634-8521, Japan; 3Education Development Center, Nara Medical University, Nara 634-8521, Japan; 4Healthcare and Wellness Division, Mitsubishi Research Institute Inc., Tokyo 100-8141, Japan; 5Department of Community-Based Medicine, Nara Medical University, Nara 634-8521, Japan; 6Department of Doctor-Patient Relationships, Nara Medical University, Nara 634-8521, Japan

**Keywords:** lactic acidosis, biguanides, diabetes mellitus, population-based database, retrospective cohort study

## Abstract

Patient data from the National Database of Health Insurance Claims and Specific Health Checkups of Japan (NDB) are used to assess the effect of biguanide administration on rates of lactic acidosis (LA) in hospitalized diabetes mellitus (DM) patients. In this retrospective cohort study (from April 2013 to March 2016), we compare DM inpatients prescribed biguanides to DM inpatients who were not prescribed biguanides to quantify the association between biguanides and incidence of LA. In total, 8,111,848 DM patient records are retrieved from the NDB. Of the 528,768 inpatients prescribed biguanides, 782 develop LA. Of the 1,967,982 inpatients not prescribed biguanides, 1310 develop LA. The rate ratio of inpatients who develop LA and are administered biguanides to those who developed LA without receiving biguanides is 1.44 (95% CI, 1.32–1.58). Incidence rates and rate ratios for both sexes are elevated in the group prescribed biguanides for patients aged 70 years and older, markedly in those 80 years and older: 40.12 and 6.31 (95% CI, 4.75–8.39), respectively, for men and 34.96 and 5.40 (95% CI, 3.91–7.46), respectively, for women. Biguanides should be used conservatively in patients older than 70 years, particularly for those with comorbidities, and with caution in patients 80 years and older.

## 1. Introduction

In their 2015 joint position statement, the American Diabetes Association (ADA) and the European Association for the Study of Diabetes (EASD) recommended initiating administration of the biguanide metformin at or soon after diagnosis of type 2 diabetes mellitus (DM) [1]. In recent years, other drugs such as SGLT2 inhibitors and GLP-1 receptor agonists have attracted attention because of their benefits for preventing cardiovascular events [2,3]. However, metformin is inexpensive, has a low risk of hypoglycemia, and can suppress cardiovascular events regardless of blood glucose levels [4]. In the 2017 Mastermind Cohort Study in the United Kingdom, metformin was used as the first-line drug in 59.8% of cases [5]. In the United States, first-line use of metformin increased from 60% in 2005 to 77% in 2016 [6]. In Taiwan, biguanides are the most commonly prescribed oral hypoglycemic agent, and there was an increase in their use from 64.3% in 2005 to 74.4% in 2012 [7]. However, according to previous research based on the National Database of Health Insurance Claims and Specific Health Checkups of Japan (NDB), the use of biguanides including metformin in Japan was remarkably low in comparison with other nations, accounting for just 28.9% of all diabetic drugs [8].

Therapeutic dosing of metformin is associated with metformin-associated lactic acidosis (LA), a serious complication of metformin treatment with mortality rates of 30 to 50 percent [9]. In the late 1970s, global reports of fatalities associated with LA caused by biguanides resulted in prescribers limiting the metformin dose to 750 mg/day in Japan, where prescription of biguanides had been strictly controlled by specialists [10]. Perhaps due to prescriber hesitancy, the prescription volume of biguanides in Japan is remarkably low compared to other nations [5,6,7,8], although a prescription amount of 2250 mg/day is allowed at present.

Numerous studies have explored whether the relationship between biguanides and LA is causal, associated, or coincidental. The incidence rate of metformin-associated LA reportedly ranges from 0 to 138 cases per 100,000 person-years [11,12,13,14,15,16,17]. A study indicated that the incidence rate of LA was higher in a diabetic group than in a non-diabetic group, but the rate was not significantly higher in patients taking metformin [18], suggesting that the risk is increased by the presence of diabetes. As the relationship between the degree of metformin accumulation and the severity of a secondary hypoxic condition, which relates to LA risk, is clinically complex [15], liver and kidney functions related to metformin metabolism and cardiovascular problems, such as a heart failure caused by secondary hypoxia, can be potential confounders associated with the risk of developing LA. However, the results of pharmacophysiological and pharmacopathological analyses of metformin have suggested that LA is rarely induced if the liver and kidneys are able to metabolize lactate [11].

Recently, clinical attention seemed to be paid to the protective effects of metformin. As for a renal-protective effect, a recent nationwide cohort study in Korea found that metformin use was associated with lowering risk of all-cause mortality and end-stage renal disease during follow-up in patients with diabetes and chronic kidney disease than in non-users [19]. Under appropriate use of metformin doses, even patients with chronic liver disease show low concentrations of plasma lactate and metformin. When the concentrations are within the safety thresholds of these substances, some studies report that patients with diabetes, chronic liver disease, or renal dysfunction should not be kept away from using metformin [20,21].

On the other hand, the recent study in patients with advanced chronic kidney disease reported that treatment with metformin was not only associated with an increased risk of cardiac and cerebrovascular events in patients with diabetes and chronic kidney disease but also may significantly increase the risk of LA, as defined by laboratory data, as the eGFR declined [19,22]. At this point, metformin is contraindicated because of an increased risk of LA in elderly patients, patients with hepatic or renal insufficiency, and patients with circulatory problems such as congestive heart failure [9].

The Japan Diabetes Society (JDS) recommended in 2012 that physicians use their clinical judgment with abundant caution when prescribing biguanides for DM patients 75 years or older [23]. This recommendation followed verification by the JDS committee of the association of biguanides with 10 serious cases of LA including fatalities, almost all among patients 75 years and older, that were reported by pharmaceutical companies in December 2011 [24]. However, Yokoyama et al., Gregorio et al., and Lin et al. have reported no significant age-related differences in plasma lactate levels of patients prescribed metformin [16,25,26]. Few large-scale clinical studies have targeted elderly diabetics in Japan, and it is not clear whether age is a risk factor for LA [12,16].

Recently, the National Database of Health Insurance Claims and Specific Health Checkups of Japan (NDB), maintained by the Ministry of Health, Labour, and Welfare, has become available to researchers, allowing total-population retrospective analyses that virtually eliminate selection bias. Containing medical records on more than 100 million individuals, the NDB contains datasets of medical care received by inpatients and outpatients in hospitals and private practices in Japan under universal health insurance coverage [27,28,29].

In Japan, one patient can have multiple insurance claim numbers because even though the Japanese insurance system is unified, single identification numbers (e.g., Social Security numbers in the United States) are not effectively utilized. An insurance identification number can change because of changes in employment or name. For example, records under a maiden and a married name could count as two persons. To address this issue and enable the tracking of individuals, an anonymous personal identification (ID) variable named identification 0 (ID 0) was developed by Nara Medical University to link individual patient claims under varied insurance ID numbers to ID0 [27,28,29].

The purpose of this study was to assess the association between administration of biguanides and risk of developing LA among hospitalized DM patients in Japan using a retrospective cohort extracted from the NDB. Using total-population data allows identification and international comparison of cases of LA, including elderly patients, who are rarely included in research studies in Japan. This large dataset permits identification and inclusion of a greater number of elderly patients than typical study populations.

## 2. Materials and Methods

### 2.1. Dataset

The National Database of Health Insurance Claims and Specific Health Checkups of Japan (NDB) is a comprehensive administrative database containing health insurance claim information. The NDB data provide information on personal identifiers, date, age group, sex, description of the procedures performed, World Health Organization International Classification of Diseases (ICD-10) diagnosis codes, medical care received medical examinations conducted without the results, prescribed drugs, etc. The structure of the NDB originally complicates long-term cohorts for two main reasons. First, the NDB data are stored on a per-claim basis not per-episode or event basis. Second, the NDB is a billing-focused record structure, which is not built for analysis. Then, a team of NaraMedilcal University modified the data structure to allow for long-term cohorts using ID0, a process that did not increase the data volume and actually shortened the runtime per data year. The NDB uses two primary keys (ID1 and ID2) derived from hash values that mask personally identifiable information. ID0 is our key developed from ID1 and ID2, which improves patient-matching efficiency with excellent long-term tracing performance. Our study used claim data with long dates between April 2013 and March 2016 to trace hospitalizations of one month or longer, including outpatient care, in three steps. In step one, claims were transferred to a CD-record format. As some diagnosis procedure combination (DPC) claim records contain a mixture of overlapping comprehensive and piece-rate data, we sorted and reorganized them. In step two, pharmacy and medical outpatient claims were integrated using the ID0 key, the medical institution code for issuing a prescription, and the prescription issue date. In step three, the transferred data were combined and converted from consecutive hospitalization days into sequences based on ID0, the medical institution code, and hospital ward classification. Consequently, the size of the originally extracted comma-separated variable dataset for three years (approximately 10.5 TB) was reduced to an approximately 6TB main database file that was usable for processing. Our technique makes it easier to perform follow-up and longitudinal cohort surveys while accurately tracing patient data in a large-scale medical claims database [30].

### 2.2. Study Design

This retrospective cohort study analyzed insurance claims data from the NDB over the three-year period from April 2013 to March 2016. A total of 109,780,160 insurance claims were identified in the NDB during the study period, excluding patients who received publicly funded health care from Public Assistance and foreigners who had stayed in Japan for less than three months [29].

### 2.3. Study Population

Records of DM patients were retrieved from the NDB using the ID0 variable [27,28,29]. DM patients were defined as patients with at least one diagnostic code indicating a DM diagnosis and who had been prescribed at least one diabetic drug, including sulfonylureas, dipeptidyl peptidase-4 inhibitors, and various types of insulin [27,28,29]. The list of pharmaceutical codes is shown in Appendix A, along with diagnostic code S2 [27,28,29]. The patients were included in the study upon admission to the hospital rather than at discharge. The study population was then limited to DM inpatients to select patients with high adherence to medication regimens under medical supervision.

### 2.4. Exposure: Biguanide Prescriptions

Exposure was defined as having a biguanide prescription in the patient record retrieved from the NDB. DM patients who were hospitalized for any reason and prescribed biguanides at least once during the study period were defined as the exposed group. These patients were compared with an unexposed group of DM patients who were hospitalized for any reason but not prescribed biguanides during the study period. Drug information for biguanides included total volume of prescription, brand name, active ingredient, dosage, and number of days prescribed. Pharmaceutical codes for biguanides are listed in Appendix A; although the vast majority of prescriptions in Japan are for metformin, we included a comprehensive list of biguanides to capture as large a sample as possible.

### 2.5. Outcome: LA Incidence in Hospitalized DM Patients Retrieved from the NDB

The primary outcome was incidence of LA among hospitalized DM patients. Two groups of participant records were identified: disease cases among the exposed group (LA cases among hospitalized DM patients prescribed biguanides) and disease cases among the unexposed group (LA cases among hospitalized patients not prescribed biguanides). If a patient’s record in the NDB included LA during the study period, that patient was considered to have developed LA in this study and counted as one case of LA; tracking of the patient record ended at the point of diagnosis. Patient records where a biguanide prescription was prescribed after the onset of LA were not included in the study.

ICD-10 code E87.2 was used to identify DM inpatients with LA diagnoses in the NDB. Of cases with this code, the Japanese diagnosis code for LA (#20072477) was retrieved and defined as LA (Figure 1).

Since the NDB does not include laboratory test data, LA patients were identified and retrieved using this diagnostic code rather than clinical criteria.

### 2.6. Statistical Analysis

The person-time method was used to calculate the incidence rate of developing LA during hospitalization because it can account for the effect of death or loss to follow-up in the patient record [31]. Person-days is the sum of the number of person-days at risk for each patient. In this case, at risk is the sum of the number of days during the observation period that the patient was insured and did not have lactic acidosis.

The calculation of the incidence rate involved in the hospitalized, exposed group was (the number of LA of exposed group/the number of person-day) × 365 days × 100,000 DM patients.
$$\frac{The\;number\;of\;LA\;(a)}{The\;number\;of\;person-day}\times 365\;\mathrm{days}\times 100,000\;\mathrm{DM}\;\mathrm{patients}$$

The calculation of the incidence rate involved in the hospitalized, unexposed group was (the number of unexposed group /the number of person-day) × 365 days × 100,000 DM patients.
$$\frac{The\;number\;of\;LA\;(c)}{The\;number\;of\;person-day}\times 365\;\mathrm{days}\times 100,000\;\mathrm{DM}\;\mathrm{patients}$$

The rate ratio of developing LA was computed by dividing the incidence rate of LA among the exposed group divided by the incidence rate of LA among the unexposed group; 95 percent confidence intervals (CI) were also computed. Stratified analyses by age and sex were also calculated to determine if rate ratios varied by these potential confounding variables. Categorical variables of five-year increments for age were created for patients aged 65 to 94 years. Whenever the number in any subgroup was fewer than 10, data were masked so that no individuals could be identified. Patients under 65 years from the unexposed group were used as a reference group. A Microsoft SQL Server 2016 was used for data analysis.

## 3. Results

The retrieved data comprised the largest number of LA patients in Japan to date, and more than half were aged 65 or older. During the three-year study period, 8,111,848 patients in the NDB were identified as having DM and having been prescribed diabetic medications. Of these, 2,344,096 had been prescribed biguanides, and 5,767,752 had not been prescribed biguanides. A total of 2,496,750 DM patients had been hospitalized for any reason. Of those prescribed biguanides (exposed), 782 had an LA diagnosis. Of those who did not receive biguanides (unexposed), 1310 had an LA diagnosis (Table 1 and Figure 1).

The incidence rate of developing LA was 13.41 versus 9.34 per 100,000 person-years in the exposed and unexposed groups, respectively, a rate ratio of 1.44 (95% CI, 1.32–1.58) (Table 2).

When stratified by age and sex, differences emerged. LA incidence rates per 100,000 person-years were 7.61 and 6.36 for male patients under 65 years old in exposed and unexposed groups, respectively, though this difference was not statistically significant (rate ratio: 1.20 (95% CI, 0.96–1.50)). For female patients under age 65, the LA incidence rate per 100,000 person-years was 9.47 and 6.48 in the exposed and unexposed groups, respectively (rate ratio: 1.46 (95% CI, 1.09–1.96)). For both sexes, incidence rate and rate ratios of developing LA increased with age starting at age 70. Comparing patients 80–84 years to those in their 60s (the reference group), the rate ratio for men was 6.31 (95% CI, 4.75–8.39) and 5.40 for women (95% CI, 3.91–7.46). Comparing patients 85–89 years to those in the reference group, the rate ratio was 4.97 for men (95% CI, 2.98–8.29) and 6.68 for women (95% CI, 4.50–9.91). Among patients 80 years and older, both rate ratio and incidence rate were more than double that of DM patients in their 70s (Table 3 and Figure 2).

For men aged 80 and older, the incidence rate of the exposed group was on average 1.5–2.5 times higher than the unexposed group, and for women 80 years and older, the incidence rate among the exposed was 2.5–3.0 times higher than the unexposed group (Table 3 and Figure 2).

## 4. Discussion

Although LA is a rare outcome, we identified and analyzed a total of 2092 cases of LA in this study from patient records in the NDB. Using the NDB reduced selection bias and allowed generation of the first total-population dataset in Japan to assess the risk of developing LA associated with biguanide use. As well as minimizing bias compared to other study designs, use of the total-population dataset identified a large number of LA patients. A substantial proportion of elderly patients who are typically underrepresented in studies evaluating biguanide use are included. In this study, the large number of LA patients allowed stratified analysis by age. This study relied on clinical codes, not clinical diagnostic criteria, yet found a rate ratio (1.44) within the range of those found by other studies (0.69 to 13.53), suggesting this alternative methodological approach still yielded plausible findings [32,33].

The LA incidence rate for inpatients exposed to biguanides was markedly higher than for the unexposed, consistent with the results of previous studies [15,16,17,34]. The genetics of diabetes audit and research (GoDartS), which used lactate and bicarbonate measurements rather than relying on the ICD code, showed that the use of metformin could increase plasma lactate concentrations and more than doubled the risk of LA with a biochemical diagnosis [17]. Although the results of our study showed that the development rate of LA was not as high as the GoDartS study [17], biguanides have been suggested to be associated with the development of LA.

Among hospitalized DM patients prescribed biguanides, the risk of developing LA was lower for younger patients than for patients aged 70 years and older. Reported LA incidence rates for patients prescribed metformin range from 3 to 10 per 100,000 person-years in countries including the USA, Canada, and a number of European nations [9,11]. In our study, the incidence rate for patients younger than 65 years fell into this range. However, incidence rates for older patients were over 10 per 100,000 person-years, and the higher rate of LA among this subgroup skewed the overall incidence rate higher (Table 3). Incidence rate and rate ratio of LA both increased significantly with patient age, suggesting that age is a risk factor for LA and could be used in DM patients to predict the risk of developing LA. This finding supports the JDS recommendation to use caution in prescribing biguanides for patients aged 75 years and older.

Metformin-associated LA usually requires both high plasma levels of metformin (often resulting from renal impairment) and a secondary condition that impairs lactate production or clearance, such as sepsis or cirrhosis [9]. Thus far, other studies have differed in their findings on the associations of LA incidence rate with any of these factors [22,25,30,32,35,36,37,38,39].

The REMIND-TMU study investigated the relationship between metformin use and lactic acidosis in advanced chronic kidney disease, type 2 diabetic adults with eGFR <30 mL/min/1.73 m^2^. Even after adjustment for confounders by age, sex, and comorbidities, metformin significantly increased the risk of lactic acidosis (*p* = 0.0204). It was concluded that this was due to a decrease in eGFR in advanced chronic kidney disease with an average follow-up of more than 600 days, but the tendency of increased risk is also consistent with the results of our study [22].

In a study by Lazarus B et al., occurrence of LA hospitalization in DM patients with reduced kidney function (estimated glomerular filtration rate (eGFR) <45–59 mL/min/1.73 m^2^ and eGFR 30–44 mL/min/1.73 m^2^) were not statistically different between patients treated with metformin and those treated with alternative diabetes management [32]. The median age of patients was 60.4 years in the study.

In a retrospective analysis of a large number of DM patients who developed metformin-associated LA in a study in Italy over a six-year period [34], the LA incidence was 12.04/100,000 in metformin-treated diabetics. The mean age in the study was 71.6 years. The findings of Lazarus B et al. and Mariano et al. indicate that patients in their 60s are less likely to experience LA with metformin use but that caution should be exercised when prescribing metformin to patients in their 70s [34]. From this, it is recommended that the patient’s age is taken into account whenever biguanide use is considered.

Of course, as was the case with the aforementioned study subjects of Lazarus B et al., renal function tends to decrease with age, and the risk of comorbid conditions increases, explaining the elevated risk of LA among elderly patients administered biguanides. Even though it is recommended that metformin be used in elderly patients with regular assessment of renal function using eGFR, eGFR calculated from serum creatinine may overestimate renal function in late elderly patients because of loss of muscle mass [40]. In patients 80 years and older, caution should be exercised in prescribing biguanides, taking drugs that reduce renal blood flow, particularly those with comorbid conditions that raise the risk of LA development.

Although this study used an NDB-derived dataset, which included almost all insured DM patients in Japan, patients whose DM was treated through non-pharmacological modalities such as diet and exercise, as well as welfare beneficiaries without insurance, were not included in the dataset because they were impossible to identify and include. The wealth of information in such a large, comprehensive dataset likely outweighed this drawback.

In addition, the actual diagnosis of LA is difficult without information that metformin was used. As reported by van Berlo-van de Laar IRF et al., the sensitivity of identifying metformin-associated LA in patients with sepsis-induced lactic acidosis suspected is 85% when even using the recommended parameters (lactate ≥ 8.4 mmol/L, creatinine ≥ 256 μmol/L), and a specificity is 95% [41]. Under this circumstance, underreporting cases should be considered. Because the LA diagnoses we used were based on clinical codes, not clinical diagnostic criteria applied to laboratory results, some bias may exist in the dataset, either toward underreporting if LA was present but not diagnosed or toward overreporting if suspected LA was tentatively diagnosed. Use of hospital data from a large comprehensive medical database and careful use of clinical disease codes to identify cases likely minimized this risk of bias, evidenced by incidence of LA comparable to reports from other countries. Limiting to research on inpatients as subjects in this study should reduce the number of suspected disease patients.

Last, we did not examine the effects of differing dosages, types of biguanides prescribed, drug-to-drug interactions, and severity of comorbidities on the incidence of LA, all potential effect modifiers that should be investigated further. According to Yokoyama et al., high doses of metformin are considered to be an independent risk factor for increased lactate levels in DM patients [16]. Xu Cheng et al. have also reported that the association between metformin use and acidosis is significantly correlated with the severity of COVID-19, in addition to the amount of metformin used [37]. The possibility of overestimating incidence ratios cannot be discounted, but cases of LA from other causes are equally included in both the biguanide prescription (exposed) and no biguanide prescription (unexposed) groups. Therefore, the rate ratio was used as the relative risk of administering biguanides in this study. A number of cases of LA caused by factors other than metformin were possibly included in this study. Taking this into consideration, the authors attempted to visualize the relative risk of biguanides by showing rate ratios, and the LA risk caused by biguanides was estimated. The influence of factors other than metformin should still be considered, however. Nevertheless, our results show the incidence rate of LA based on real-world data in the biguanide and non-biguanide prescription groups, and the findings in this study should provide insights and be useful in clinical practice from the viewpoint of descriptive epidemiology.

Furthermore, as potential confounding factors on the relationship between age and risk of developing LA after biguanide use, obesity, heart, and renal and/or liver function should be considered, though data on these factors were not available in this study [16,22,25,26,32,33,36,37,38,39,42]. NDB data available to researchers included neither anthropometric information (e.g., height and weight) nor patient laboratory test results, so we were unable to ascertain whether any of these factors affected the association between biguanide prescription and LA incidence rate. If researchers are eventually allowed access to laboratory results and anthropometric information in the NDB (currently not permitted), a future study could clarify the contribution of these other clinical factors to LA development.

## 5. Conclusions

This is the first retrospective cohort study to use the total-population database (NDB) of health insurance claims in Japan to examine the association of biguanide use with risk of LA in hospitalized Japanese DM patients. Both the incidence rate and rate ratio of developing LA were markedly higher among hospitalized DM patients who were prescribed biguanides than those who were not, regardless of sex. Older age groups were at significantly elevated risk of developing LA, suggesting that age should be considered a risk factor for developing LA in DM patients. It suggests that biguanides should only be prescribed after carefully weighing age-specific risks against potential benefits given an individual’s medical history and comorbidities, such as conservative use in patients over 70 years of age, especially those with comorbidities, and cautious use in patients over 80.

## Figures and Tables

**Figure 1 ijerph-20-05300-f001:**
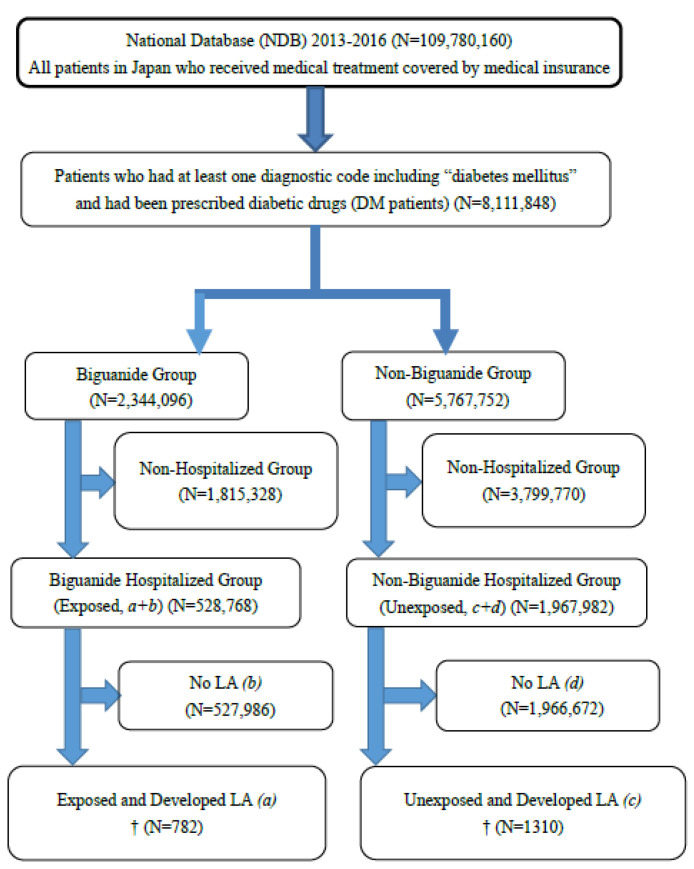
Retrieval Flow Chart for DM and LA Patients. † Acidosis was retrieved using ICD-10 code E87.2, and only LA was extracted using the disease code 20072477.

**Figure 2 ijerph-20-05300-f002:**
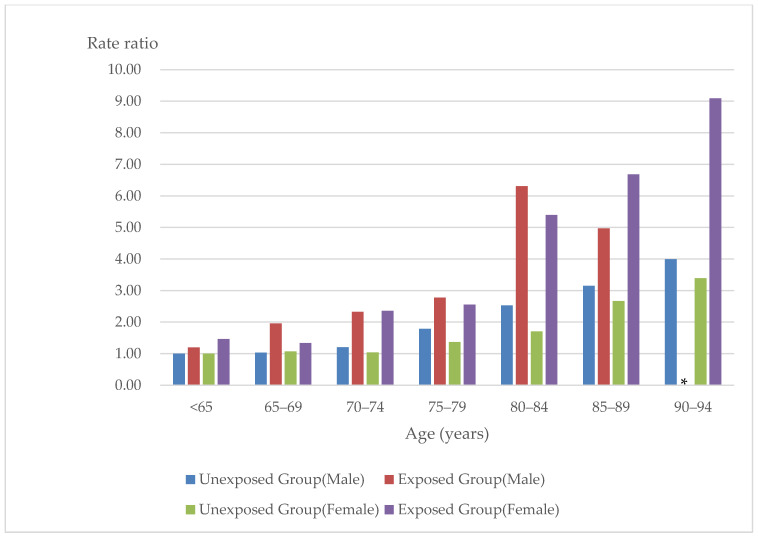
Rate Ratio of Lactic Acidosis by Age among Hospitalized DM Patients in Japan. * Denotes masking where there were fewer than 10 patients.

**Table 1 ijerph-20-05300-t001:** Exposure to biguanides and LA outcomes among hospitalized DM patients.

Hospitalized DM Patients	Lactic Acidosis Diagnosis (Disease)	No Lactic Acidosis Diagnosis (No Disease)	Total
Biguanide prescription (Exposed)	782 (a)	527,986 (b)	528,768 (a + b)
No biguanide prescription (Unexposed)	1310 (c)	1,966,672 (d)	1,967,982 (c + d)
Total	2092 (a + c)	2,494,658 (b + d)	2,496,750 (a + b + c + d)

**Table 2 ijerph-20-05300-t002:** Patient Group Classification, Incidence Rates, and Rate Ratio.

DM Patients	Number of Person-Days at Risk of LA	Patients	Incidence Rate of LA ^†^	Rate Ratio of Developing LA ^‡^ (95% CI)
Biguanide prescription (Exposed)	2,128,760,399 person-days	Diagnosed with LA (a) (N = 782)	13.41 /100,000 person-years	1.44 (1.32–1.58)
		Not diagnosed with LA (b) (N = 527,986)		
No biguanide prescription (Unexposed)	5,118,605,720 person-days	Diagnosed with LA (c) (N = 1310)	9.34 /100,000 person-years	
		Not diagnosed with LA (d) (N = 1,966,672)		

^†^ Incidence rate of LA: (number of LA/number of person-days at risk of LA) × 365 days × 100,000 DM patients. ^‡^ Rate ratio: incidence rate of LA in Exposed Group/incidence rate of LA in Unexposed Group.

**Table 3 ijerph-20-05300-t003:** Incidence Rate of Lactic Acidosis by Age among Hospitalized DM Patients in Japan.

	Biguanide-LA-Hospitalized Group			Non-Biguanide-LA-Hospitalized Group		
	(Exposed Group)			(Unexposed Group)		
Age (years)	Incidence Rate /100,000 Person-Years	Rate Ratio (95% CI)	Number	Incidence Rate /100,000 Person-Years	Rate Ratio (95% CI)	Number
	**Males**					
<65	7.61	1.20 (0.96–1.50)	140	6.36	Reference	182
65–69	**12.47**	**1.96** (**1.50**–**2.56**)	78	6.55	1.03 (0.80–1.32)	91
70–74	**14.79**	**2.32** (**1.76**–**3.06**)	70	7.66	1.20 (0.94–1.53)	103
75–79	**17.66**	**2.78** (**2.06**–**3.74**)	57	**11.39**	**1.79** (**1.43**–**2.23**)	137
80–84	**40.12**	**6.31** (**4.75**–**8.39**)	64	**16.08**	**2.53** (**2.03**–**3.16**)	137
85–89	**31.62**	**4.97** (**2.98**–**8.29**)	16	**20.04**	**3.15** (**2.41**–**4.11**)	77
90–94	^†^	^†^	^‡^	**25.41**	**3.99** (**2.59**–**6.16**)	23
≥95	^†^	^†^	^‡^	^†^	^†^	^‡^
	**Females**					
<65	**9.47**	**1.46** (**1.09**–**1.96**)	89	6.48	Reference	90
65–69	8.64	1.33 (0.91–1.94)	38	6.93	1.07 (0.77–1.49)	58
70–74	**15.26**	**2.36** (**1.70**–**3.27**)	60	6.73	1.04 (0.75–1.43)	64
75–79	**16.54**	**2.55** (**1.80**–**3.61**)	49	**8.87**	**1.37** (**1.02**–**1.84**)	87
80–84	**34.96**	**5.40** (**3.91**–**7.46**)	62	**11.02**	**1.70** (**1.28**–**2.27**)	96
85–89	**43.29**	**6.68** (**4.50**–**9.91**)	34	**17.29**	**2.67** (**2.01**–**3.55**)	98
90–94	**58.9**	**9.09** (**5.08**–**16.26**)	13	**21.99**	**3.39** (**2.40**–**4.78**)	51
≥95	^†^	^†^	^‡^	^†^	^†^	^‡^

^†^ Incidence rate and 95% CI not calculated due to masking data. ^‡^ Denotes masking where there were fewer than 10 patients. **Bold**: *p* < 0.01.

## Data Availability

Data analyzed in this study cannot be shared because NDB data may only be accessed by authorized individuals.

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
