# Peer review of "Risk of Lactic Acidosis in Hospitalized Diabetic Patients Prescribed Biguanides in Japan: A Retrospective Total-Population Cohort Study"

_ijerph, 2023, doi:10.3390/ijerph20075300_

Round 1

Reviewer 1 Report

The article is devoted to a very topical topic - the effect of biguanides on the development of lactic acidosis in elderly patients with diabetes mellitus. The study is population-based, the material is large, the analysis is retrospective.

I really liked the article. There are some small notes:

1. There are some stylistic mistakes in the article and it needs to be corrected with the help of a professional English translator.

2. There are only 4 sources of the last 3 years in the list of references. It is necessary to strengthen the article with sources of literature of the last 3 years.

Reviewer 2 Report

Dear authors,

We appreciate your efforts to provide the results of a population-based study, the National Database of Health Insurance Claims and Specific Health Checkups of Japan (NDB), which reveals the occurrence of Lactic acidosis (LA) in Japanese diabetic inpatients using biguanides from 2013 to 2016. However, the results shown seem doubtful. We would suggest the author re-start the study design, and think it over completely to prevent misleading results and inferences. 

1.     Table 2. 

   i.     In Footnote, the formula of the Incidence rate of LA is shown as: (Number of hospitalizations/number of person-days) x 365 days x 100,000 DM patients, which is inconsistent with the calculation results in the table. It is difficult for us to identify which item the numerator is.

 ii.    If the numerator of Exposed Diagnosed with LA is 782, then you should not use the number of person-days as the denominator, so is it in the Unexposed group.

iii. According to the use of metformin or not, the LA cases of DM patients were divided into 2 groups: the exposed and non-exposed. However, LA can be attributed to several factors such as the severity of the disease, the dose of metformin, the duration of metformin use, and the function of the patient's heart, liver, and kidney, etc. LA cases resulting from non-metformin use were not excluded in the study, indicating an overestimation of the incidence of metformin-related LA.

2.     Table 3.

i.      The results in Table 3 are not available because the incidence rate of LA is incorrect in nature.

ii    With age stratification, the authors decided to compare the incidence rate ratio between the exposed and Non-exposed groups, but only the incidence rate of the Non-exposed group <65 y/o was used as a reference. Do you have any reason for that decision? 

3.     Figure 2.

It is recommended that the author mark the title of the abscissa and ordinate.

Reviewer 3 Report

Thank you for giving me the opportunity to review this article. Could you please elaborate on the novelty aspects of your work? This will help to enhance the significance and relevance of your research beyond the local/regional context, towards a broader audience. The study included data on suspected cases of LA that may not have been confirmed, and it would be useful to note that the patients were included into the study upon admission to the hospital rather than at discharge. Adding results on the aspects of multimorbidity among patients with LA who use metformin, rather than just metformin users, would be useful. Additionally, data on the use of other antihyperglycemic medications in patients with LA who use metformin should be included, or the discussion should cover drug-to-drug interactions. Rewriting the abstract in the format of 'background, methods, results, conclusions' would increase its value. The conclusions presented in the abstract and article should be clear and consistent.

Round 2

Reviewer 2 Report

1. The authors have explained the calculation method for the incidence rate and provided additional information on the limitations of the study regarding other possible causes of lactic acidosis. Although there were no issues with data collection and study design, the authors should clarify in the manuscript the definition of "Number of person-days at risk of LA" and the source of the reported numbers (2,128,760,399; 5,118,605,720).

2. Although lactic acidosis is a known adverse reaction of biguanides, this study provides a more specific understanding that the use of biguanides is indeed associated with higher incidence rates and risk ratios of lactic acidosis in hospitalized diabetic patients. This highlights the need for clinical practitioners to be more cautious when prescribing biguanides, particularly for older patients. Therefore, it is suggested that the authors include corresponding arguments in the conclusion section, as stated in the abstract, that "Biguanides should be used conservatively in patients older than 70 years, …… and with caution in patients 80 years and older."
